# Isolation and Characterization of a Novel Lytic Bacteriophage against the K2 Capsule-Expressing Hypervirulent *Klebsiella pneumoniae* Strain 52145, and Identification of Its Functional Depolymerase

**DOI:** 10.3390/microorganisms9030650

**Published:** 2021-03-21

**Authors:** Botond Zsombor Pertics, Alysia Cox, Adrienn Nyúl, Nóra Szamek, Tamás Kovács, György Schneider

**Affiliations:** 1Department of Medical Microbiology and Immunology, Medical School, University of Pécs, Szigeti st. 12., H-7624 Pécs, Hungary; pertics.botond@gmail.com (B.Z.P.); nyul.adrienn@pte.hu (A.N.); 2Department of Biotechnology, Nanophagetherapy Center, Enviroinvest Corporation, Kertváros st. 2., H-7632 Pécs, Hungary; alcox@tcd.ie (A.C.); szamek.nora@enviroinvest.hu (N.S.); kovacst@enviroinvest.hu (T.K.)

**Keywords:** *Klebsiella pneumoniae*, K2 serotype, bacteriophage, tail fibers, capsule depolymerase

## Abstract

*Klebsiella pneumoniae* is among the leading bacteria that cause nosocomial infections. The capsule of this Gram-negative bacterium is a dominant virulence factor, with a prominent role in defense and biofilm formation. Bacteriophages, which are specific for one bacterial strain and its capsule type, can evoke the lysis of bacterial cells, aided by polysaccharide depolymerase enzymes. In this study, we isolated and characterized a bacteriophage against the nosocomial *K. pneumoniae* 52145 strain with K2 capsular serotype. The phage showed a narrow host range and stable lytic activity, even when exposed to different temperatures or detergents. Preventive effect of the phage in a nasal colonization model was investigated in vivo. Phlyogenetic analysis showed that the newly isolated *Klebsiella* phage B1 belongs to the *Webervirus* genus in *Drexlerviridae* family. We identified the location of the capsule depolymerase gene of the new phage, which was amplified, cloned, expressed, and purified. The efficacy of the recombinant B1dep depolymerase was tested by spotting on *K. pneumoniae* strains and it was confirmed that the extract lowers the thickness of the bacterium lawn as it degrades the protective capsule on bacterial cells. As *K. pneumoniae* strains possessing the K2 serotype have epidemiological importance, the B1 phage and its depolymerase are promising candidates for use as possible antimicrobial agents.

## 1. Introduction

*Klebsiella pneumoniae* is a Gram-negative, encapsulated bacterium and one of the most important opportunistic nosocomial pathogens [1]. It is ubiquitous in the environment [2] and colonizes the human skin and mucosal surfaces including the gastrointestinal tract and oropharynx, where it can be a source of serious infection of the urinary and respiratory tracts, surgical sites, and catheter entry points. These infections can progress to potentially life-threatening conditions and septicemia [1,3], most often in hospitalized and immune-compromised patients [4,5]. Community-acquired infections, such as pyogenic liver abscesses (PLA), metastatic meningitis, and endophthalmitis [1], are also reported in young and healthy individuals, contributing to a serious global public health concern in the past few decades. *K. pneumoniae* is a member of the ESKAPE (Enterococcus faecium, Staphylococcus aureus, Klebsiella pneumoniae, Acinetobacter baumannii, Pseudomonas aeruginosa and Enterobacter spp.) group of microorganisms [6] as isolates are frequently resistant to multiple antibiotics.

Capsule, the outermost layer on this pathogen, is a crucial virulence factor involved mainly in resistance to phagocytosis by acting as a physical barrier against some antibiotics and host immune mechanisms [7]. Capsular polysaccharides (CPS) can be classified into 78 serological types (K antigens) according to the structural differences in the polysaccharide chains [8]. Virulence depends on capsule type, thickness, and glycan structure [9]. Overproduction of CPS relates to mucoid phenotype, which may facilitate colonization of mucoid surfaces [10]. Although hypermucoviscosity does not necessarily induce hypervirulence [11], both traits are generally predominant in capsular types K1 and K2, the serotypes that are most frequently isolated from patients worldwide [5,10,12,13,14,15,16,17]. The resistance of the K1 or K2 capsule to phagocytosis and intracellular killing contributes to their high prevalence in liver abscesses and endophthalmitis, as K1/K2 isolates compared to others are generally more virulent in both liver abscess and in non-liver abscess conditions [18].

As antibiotic resistance is growing among the isolates, novel approaches must be considered for therapy and prevention. Bacteriophages (phages) are bacterial viruses that are able kill the target bacterial cell. Unique features of phages allow them to recognize specific receptor structures on the surface of the bacterial cells, dock and infect the host cell, subsequently releasing phage progenies [19]. Phages can effectively control bacteria enveloped with polysaccharide capsules.

The application of bacteriophages against *K. pneumoniae* have been implemented successfully in vitro and in vivo [20,21,22,23,24,25,26,27,28,29,30,31]. In order to infect capsulated *K. pneumoniae* strains, phages must be able to cross the capsule layer prior to docking and penetration. For this purpose, phages have specific polysaccharide depolymerase enzymes, which degrade the polysaccharide capsule structure, thus allowing the phage to reach the bacterial cell surface, adsorb to the outer membrane receptor, and infect the cell, followed by lysis and phage progeny release [32,33,34,35]. Capsule depolymerases have specific effects and selectivity to certain serotypes [36], though only one has been described as effective against the K2 serotype [37].

In this study, we isolated a novel lytic bacteriophage (B1) able to lyse the hypervirulent *K. pneumoniae* strain 52145 possessing the K2 serotype. Lytic activity was evaluated in functional tests and the genetic structure of B1 was determined by sequence analysis. Its preventive and therapeutic efficacy was evaluated in a murine nasal infection model. Following this, the potential capsule depolymerase coding gene of B1 was identified by cloning and expression. Its activity was confirmed by spot testing on the 52145 strain. This is the first report of a recombinant, K2 serotype specific depolymerase from a phage belonging to the *Siphoviridae* family.

## 2. Materials and Methods

### 2.1. Bacterial Strains and Growth Conditions

*Klebsiella pneumoniae* human isolate 52145 wild type (WT) and its isogenic mutants were used in this study (Table 1). Twenty strains/isolates with different capsular serotypes were also tested. Bacteria were routinely grown on lysogeny broth agar (LBA) plates at 37 °C or liquid lysogeny broth (LB) medium (37 °C at 125 rpm). To establish a bacterial lawn, we plated 100 μL of the overnight (ON) liquid cultures onto a solid LB agar plate and incubated ON at 37 °C. Bacteria were proliferated ON in liquid medium at 37 °C in an orbital shaker (125 rpm).

### 2.2. Phage Isolation, Propagation, and Titer Determination

Bacteriophages were isolated in the spring of 2016 from a local sewage farm (Pellérd, Hungary) with the traditional method [38]. Briefly, 1 mL of sewage sample was incubated with a 50 mL mid-log suspension (optical density OD_600_ = 0.5–0.6) of isolate 52145 WT ON at 37 °C. The suspension was centrifuged (4000 rpm, 10 min) and the supernatant was treated with chloroform at a 1:50 *v*/*v* ratio (Molar Chemicals Kft., Halásztelek, Hungary) ON at 4 °C. Presence of lytic phages was confirmed by spot testing [38] on the lawn of isolate 52145 WT. Spot testing was performed throughout this study as follows, unless indicated otherwise: after plating 100 µL bacteria on LB agar, 10 µL of the tested suspension was dropped, left to dry, and then incubated ON at 37 °C. A single phage plaque was cut out using the agar overlay method [39] and was purified in 3 consecutive steps. The purified phage clone, named “*Klebsiella* phage B1” according to the recent phage nomenclature [40], was propagated in 100 mL LB medium, centrifuged (11,000 rpm, 30 min), and resuspended in 50 mL deionized water (DW). The phage titers were obtained by serially diluting the phage suspensions and subsequently spotting 10 µL from each dilution to obtain a countable number of individual plaques on the lawn. Then, the number of plaque-forming units (PFU) was calculated for 1 mL of the concentrated suspension. The resulting high titer suspension (10^9^ PFU/mL) was used for further studies. “*Klebsiella* phage 731” was isolated the same way on the 52145 capsule mutant (ΔK), and furthermore it was selected to be ineffective on the wild type (WT).

### 2.3. Host Range Determination and Efficiency of Plating (EOP)

Host range of B1 phage was determined by spot testing on 24 *K. pneumoniae* isolates possessing known capsule serotypes (Table 1). Spots were recorded as (i) clear: phage can kill the bacteria; (ii) turbid: no lysis but capsule depolymerase was active; (iii) absent: the isolate is resistant to the phage. Clear spots were observed either solely or surrounded by a turbid ring (halo), with the latter indicating lysis and depolymerase activity as well.

*K. pneumoniae* isolates sensitive to phage B1 in spot test were selected for EOP determination as previously described [30], with some modifications. Briefly, the selected isolates were grown overnight at 37 °C and 100 µL of them were plated. PFU of B1 phage was enumerated by spotting the serial dilution of the respective phage suspension on the target lawn. The EOP was calculated (average PFU on target bacteria/average PFU on host bacteria). The phage efficiency was classified as highly productive (EOP ≥ 0.5), moderately productive (0.1 ≤ EOP < 0.5), low productive (0.001 < EOP < 0.1), or inefficient (EOP ≤ 0.001). Results were reported as the mean of 3 independent measurements.

### 2.4. Phage Resistance Detection, MALDI-TOF MS

Emergence of resistant mutants against phage B1 was tested on the lawn (10^7^ colony-forming unit (CFU)/100 μL) of the 52145 WT strain by dropping 10 μL dense phage suspension (10^9^ PFU/mL) on the lawn and letting it dry. After 24 h incubation, the emergence of phage-resistant colonies was visually detected. Colony-forming units (CFUs) were counted in the clear spot, and their theoretical number was extrapolated for the whole surface of the lawn. The mutation rate was calculated as the ratio between the resistant mutant CFUs and the whole bacterial CFUs of the lawn per 24 h. Results are the mean of 3 different observations. To verify that the resistant colonies were *K. pneumoniae*, we performed species confirmation by using matrix-assisted laser desorption/ionization–time of flight mass spectrometry (MALDI-TOF MS) (Vitek MS, Biomerieux, Marcy-l’Étoile, France) [41].

To determine how the colonies in the clear spots had become resistant, we used a verification method as previously described [28], with modifications. Briefly, 2 mL of log-phase 52145 WT bacteria (10^8^ CFU/mL) was co-incubated with B1 phage (10^8^ PFU/mL). The cultures were incubated for 24 h at 37 °C with shaking. Then, the suspensions were diluted 10,000×, and 100 µL from this dilution was spread on LBA plates. Single colonies after ON incubation at 37 °C were harvested by a sterile bacteriological loop, spread on LBA, and spot tested for both B1 and 731 phages.

### 2.5. One-Step Phage Growth Curve, Adsorption Assay, and Killing Efficacy

The one-step phage growth curve and the burst-size (average number of phages released by each bacterium after lysis) were determined as described previously [28], with modifications. Briefly, the host strain was grown at 37 °C until log phase (OD_600_ = 0.5–0.6). The suspension (0.9 mL; 10^8^ CFU/mL) was mixed with 0.1 mL of the phage suspension (10^7^ PFU/mL) in order to achieve a multiplication of infection (MOI) of 0.01. The mixture was incubated for 10 min at 37 °C and centrifuged at 13,000 rpm for 4 min. The pellet was resuspended in 1 mL LB and the centrifugation step was repeated in order to exclude any non-adsorbed phages from the medium. The pellet was resuspended in 1 mL LB, diluted 1:10,000 in 50 mL LB medium, and incubated at 37 °C with shaking. Aliquots of 500 µL were sampled from zero time to 1 h with 3 min intervals. Each aliquot was treated with 1:50 *v/v* chloroform (Molar Chemicals Kft., Halásztelek, Hungary) and incubated at 4 °C overnight. Samples were centrifuged (12,000 rpm, 1 min), and the PFU of the supernatant was determined by spot testing on LBA plates. The latency period was defined as the time between infection and the shortest incubation time, allowing for the production of phages. The burst size was calculated as the ratio between the number of phage particles released at the plateau level and the initial number of infected bacterial cells. The experiment was performed 3 times, and the reported values are the mean of the observations.

Phage adsorption assay was performed as described previously [42], with minor modifications. Briefly, 1 mL of exponential-phase culture (10^8^ CFU/mL) was mixed with 10 µL of diluted phage (10^7^ PFU/mL) suspension (MOI = 0.01). The mixture was incubated at 37 °C for 10 min, then centrifuged at 13,000 rpm for 5 min. The titer of the supernatant was carefully determined by spot test. The phage adsorption efficiency was calculated with the equation (initial phage titer—residual titer in the supernatant)/initial phage titer.

Lytic ability of B1 phage was demonstrated on exponential growth-phase cultures of *K. pneumoniae* 52145 WT isolate, as described previously [30]. Briefly, a 96-well tissue culture plate (Sarstedt, Nümbrecht, Germany) was partitioned into parallel sections, each containing 3 identical columns. Each row contained 180 µL bacterial culture (10^3^ CFU/mL). LB without bacteria served as a negative control and inoculated bacteria without phage served as positive control. To all other rows, 20 µL phage suspension was added with different phage dilutions: 10^7^ (MOI = 10000), 10^6^ (MOI = 1000), 10^5^ (MOI = 100), 10^4^ (MOI = 10), 10^3^ (MOI = 1), 10^2^ (MOI = 0.1), 10 (MOI = 0.01). The plate was incubated at 37 °C for 24 h shaking at 180 rpm. The growth was monitored in a Synergy HT multimode reader (BioTek, Winooski, VT, USA). The OD_600_ was measured every 5 min. Experiments were performed in 3 replicates and reproduced in three independent trials.

### 2.6. Heat and Detergent Tolerance of Phage B1

As the B1 phage showed potential as an antibacterial agent against a hypervirulent strain, practical aspects were considered to examine its durability and tenacity. In order to test its adaptation limits, we exposed B1 phage to different physical conditions.

For heat-resistance tests, 1 mL samples of the concentrated phage suspension (10^9^ PFU/mL) in LB medium were incubated at different temperatures. Aliquots were put into a cooling–heating dry block (CH-100, Biosan, Riga, Latvia) at 50, 60, or 70 °C for 1 h. Samples were taken for titer determination at 0, 10, 20, 40, and 60 min. For 23, 37, and 42 °C measurements, 1 mL suspensions were added to thermostats with the corresponding temperatures, and in a freezer for −20 °C measurements. Control samples were incubated at 4 °C or −80 °C (for the latter, 1:3 *v*/*v* 85% glycerol was added). The titer was checked after 0 min, 5 and 12 days, and 2 and 9 months.

Plaque freezing was also performed to test if B1 phage can tolerate this long-term storage method [43]. The suspension (10 µL; 10^9^ PFU/mL) was dropped onto 3 different places of LB agar plates with 52145 lawn. After ON incubation at 37 °C, approximately 0.5 cm^3^ agar cubes were excised with a sterile scalpel (≈10^7^ PFU/mL). Three cubes were added to a 1.5 mL microcentrifuge tube, which was centrifuged (3–5 s) and stored at −20 °C. Tubes were thawed at room temperature after 5 and 18 days. LB broth (200 µL) was added, followed by vigorous vortexing. Samples were then titered on 52145 lawn by spot test.

B1 phage was also exposed to different detergents: Tween 20, Tween 80, Triton X-100, 10% sodium dodecyl sulfate (SDS), and DW as control. Concentrated phage suspension (10 µL; 10^9^ PFU/mL) was added to 990 µL detergent. Samples were stored at 4 °C and titered after 1 h, 1 day, 1 week, 1 month, and 9 months.

All experiments were carried out in triplicate, and results were obtained as the mean of 3 measurements.

### 2.7. Transmission Electron Microscopy (TEM)

Morphology of the phages was performed by TEM as described recently [30]. Briefly, 10 µL from the purified high titer (10^9^ PFU/mL) phage stock was deposited onto formvar-coated copper grids (Pelco Grids, Redding, Canada) and negatively stained with 1.5% *w/v* phospho-tungstic acid (Merck KGaA, Darmstadt, Germany) for 40 s. After drying, phages were visualized on a JEM-1400 Flash TEM (JEOL USA Inc., Peabody, MA, USA) operated at 80 kV acceleration voltage, with 54 µA beam current.

### 2.8. Phage DNA Extraction, Genome Sequence, and Bioinformatic Analysis

DNA was extracted from 1.5 mL phage suspensions with titers of 10^9^ PFU/mL. Suspensions were centrifuged for 10 min at 10,000 rpm. Then, 10 μL DNase I (1 mg/mL D4527, Sigma-Aldrich, dissolved in 0.15 M NaCl) and ≈10 mg RNase A crystals (R5503, Sigma-Aldrich, St. Louis, MI, USA) were added to the supernatant and incubated for 30 min at room temperature. After incubation with 100 μL 0.5 M EDTA (pH 8) for 10 min at 75 °C, 20 μL Proteinase K (10 mg/mL, recombinant, PCR Grade, 03 115 852 001, Roche Diagnostics GmbH, Basel, Switzerland) was added, followed by 1 h incubation at 65 °C. After supplementing with 50 μL 7.5 M ammonium-acetate, phage DNA was extracted with phenol–chloroform (phenol/chloroform/isoamyl alcohol 25:24:1 *v*/*v*/*v*, saturated with 10 mM Tris (pH 8.0) and 1 mM EDTA, P2069, Sigma-Aldrich, St. Louis, MI, USA) and precipitated with 96% ethanol.

The purified phage DNA was dissolved in 100 μL of sterile nuclease-free H_2_O and was used to prepare genomic DNA sequencing libraries by using the Nextera XT Library Preparation kit (Illumina, San Diego, CA, USA). Sequencing was performed using MiSeq Reagent Kit v2 (2 × 150 bp) on an Illumina MiSeq instrument (Illumina, San Diego, CA, USA). The Mypro pipeline was used to assemble the gained pure sequences.

The assembled sequence was annotated on the RAST server (https://rast.nmpdr.org/, access date: 19 March 2021). CLC Sequence Viewer v.6 (CLC bio, Aarhus, Denmark) was used to analyze the annotated genome and Easyfig 2.2.2 (https://mjsull.github.io/Easyfig/, access date: 19 March 2021) to illustrate the genome map. Open reading frames (ORFs) and gene predictions were confirmed by GeneMarkS program [44]. Genome was searched for restriction endonuclease recognition sites in silico by Webcutter 2.0 online program (http://heimanlab.com/cut2.html, access date: 19 March 2021). Homology searches were conducted by the BLAST tools available at the NCBI website (https://blast.ncbi.nlm.nih.gov/Blast.cgi, access date: 19 March 2021). Phage was classified according to the guidelines of the International Committee on Taxonomy of Viruses (ICTV, talk.ictvonline.org/taxonomy/, access date: 19 March 2021) supported by ViralZone (viralzone.expasy.org, access date: 19 March 2021) and BLASTn results. Protein homology and conserved domain prediction was conducted by NCBI BLASTp, InterProScan (http://www.ebi.ac.uk/Tools/pfa/iprscan/, access date: 19 March 2021) and HHpred tool of MPI Bioinformatics Toolkit (toolkit.tuebingen.mpg.de/tools/hhpred, access date: 19 March 2021).

The nucleotide sequence of B1 phage was deposited in the GenBank database under the accession number MW672037.

### 2.9. Phylogenetic Analysis of Phage B1

Whole genome-based phylogenetic analysis was conducted with VICTOR [45], involving the first 30 highly similar *Klebsiella* phages, according to the homology searches, and our *Klebsiella* phages. All pairwise comparisons of the nucleotide sequence were conducted using the genome–BLAST distance phylogeny (GBDP) method, under settings recommended for prokaryotic viruses. Branch support was inferred from 100 pseudo-bootstrap replicates each. The tree was rooted at the midpoint [46] and visualized with FigTree [47]. Taxon boundaries at the species, genus, and family levels were estimated with the OPTSIL program, with the recommended clustering thresholds and an F value (fraction of links required for cluster fusion) of 0.5 [48].

### 2.10. Cloning of the Putative Depolymerase of the B1 Phage

The designed primer pairs containing overhanging ends with *Xho*I and *Eco*RI enzyme recognition sites (5′–TAAATGTTACTCGAGATGGCACTATACAGAGAAGGCAAAG–3′ and 5′–TAAATGTTAGAATTCCTTAGCTACTCATAAATCCATTTGCTTTCAAAG–3′ (Sigma-Aldrich, St. Louis, MI, USA)) were used to amplify *orf61* from the purified DNA of B1 phage, coding for a tail fiber with putative depolymerase motif. The PCR reaction was performed by using the PFU DNA polymerase enzyme mix (Thermo Fisher Scientific, Waltham, MA, USA) with the following conditions: denaturation at 95 °C for 2 min followed by 30 cycles consisting of 95 °C 30 s denaturation, 54 °C 30 s annealing, and 72 °C 1 min elongation. The amplification was ended by an additional 10 min 72 °C post-elongation phase. The amplified PCR fragment was separated (1% agarose gel), extracted (Gel Extraction Kit; Thermo Fisher Scientific, Waltham, MA, USA), and blunt-end cloned (ON, 20 °C) into the linearized pJET1.2/blunt plasmid by using the Clone JET PCR Cloning Kit (Thermo Fisher Scientific, Waltham, MA, USA). The ligation mixture was heat shock-transformed into the laboratory *Escherichia coli* strain XL1-Blue. The integrated PCR insert was double-digested with *Xho*I and *Eco*RI (Thermo Fisher Scientific, Waltham, MA, USA), and after separation (1% agarose gel) and the consecutive purification (Gel Extraction Kit) step, it was re-ligated into the *Xho*I and *Eco*RI linearized pRSET A expression vector (Thermo Fisher Scientific, Waltham, MA, USA). Ligated constructs were ethanol-precipitated and after resuspension in DW, and were electroporated into the laboratory *E. coli* strain KRX (Promega, Madison, WI, USA) by using 2 mm diameter cuvettes and the GenePulser XCell system (Bio-Rad, Hercules, CA, USA) with 1.8 kV voltage and 800 Ω resistance. Transformed cells were selected on ampicillin-containing (100 µg/mL) LB agar plates incubated ON at 37 °C. Integration and presence of the resulting *orf61*-pRSET A construct was confirmed with the double-digestion of the isolated plasmids and with PCR.

### 2.11. Depolymerase Activity Testing and 1 Dimensional Gel-Electrophoresis

ON cultures (5 mL) of (1) KRX, (2) KRX with pRSET A, and (3) KRX containing the *orf61*-pRSET A plasmid construct were used to start log phase (OD_600_ = 0.8) cultures. Cultures were incubated without (control) or with isopropyl-β-D-thiogalactoside (IPTG) at 2 mM final concentration. From the resulting cultures, 1 mL samples were taken at 0, 1, 2, 3, 4, and 5 h timepoints. Samples were centrifuged and the pellets were re-suspended either in DW boiling at 100 °C for 10 min or in sonication buffer (pH 7.4; 50 mM Tris, 1 mM EDTA) for sonication. Sonication was performed for 4 cycles of 0.5–0.5 s of pulsation for 1 min at 25–30% amplitude. After each cycle, there was a 1–2 min incubation period on ice. Boiled or sonicated samples were mixed with 1:4 (*v*/*v*) volume of 5× Sample Buffer (0.6 mL 1 M Tris (pH 6.8), 5 mL 50% glycerol, 2 mL 10% SDS, 0.5 mL β-mercaptoethanol, bromophenol blue, filled with DW up to 10 mL) and electrophoresed on a 10% polyacrylamide gel [49] for 1.5 h at 120 V, using the Bio-Rad Mini Protean II system (Bio-Rad, Hercules, CA, USA). After separation, bands were visualized with Coomassie Brillant Blue (R-250, Reanal, Budapest, Hungary) staining [50].

Polyhistidine tag-based affinity purification of the expressed depolymerase was performed from 5 mL log phase culture that was induced with IPTG (2 mM) for 5 h. The centrifuged pellet was re-suspended in His-binding buffer and was sonicated as described above. Purification of the protein was implemented with the His-Spin Protein Miniprep (Zymo Research, Irvine, CA, USA), according to the protocol. The purified protein was also visualized by polyacrylamide gel electrophoresis.

Capsule degrading activity of the purified recombinant depolymerase was assessed by spot testing on the lawn of different *K. pneumoniae* strains.

**Table 1 microorganisms-09-00650-t001:** Bacterial strains used in this study, and effect of B1 phage obtained by spot testing. Results are the mean of three different experiments. +: clear lysis, −: no effect, *H*: turbid ring around lysis (halo), H: turbid spot without lysis. The phage efficiency was classified as highly productive (efficiency of plating (EOP) ≥ 0.5), moderately productive (0.1 ≤ EOP < 0.5), low productive (0.001 < EOP < 0.1), or inefficient (EOP ≤ 0.001).

*K. pneumoniae* Strain/Isolate, Reference/Source	Serotype	Phage B1	
		Visible Effect	EOP
52145 [51]	O1:K2	+*H*	High
52145-Δ*wca*_K2_ [52]	O1:K–	-	Inefficient
52145-Δ*waaL* [53]	O–:K2	+*H*	High
52145-Δ*wca*_K2_Δ*waaL* [2]	O–:K–	+	Low
NTUH-K2044 [54]	K1	-	Inefficient
CIP 52.145	K2	+*H*	High
CIP 80.51	K3	-	Inefficient
ATCC 700603 [55]	K6	-	Inefficient
CIP 52.207	K9	-	Inefficient
CIP 52.215	K11:O3	-	Inefficient
CIP 52.216	K12:O1	-	Inefficient
CIP 52.217	K13	H	Inefficient
CIP 52.221	K17	-	Inefficient
CIP 52.223	K19	-	Inefficient
CIP 52.224	K20	-	Inefficient
CIP 52.225	K21	-	Inefficient
53/3 [30]	K24	-	Inefficient
CIP 52.229	K24	-	Inefficient
CIP 52.232	K27:O2	-	Inefficient
CIP 52.235	K30:O1	-	Inefficient
CIP 53.8	K33:O3	-	Inefficient
CIP 53.23	K47:O1	-	Inefficient
MGH 78578 [55]	K52	-	Inefficient
CIP 80.47	K64	-	Inefficient

### 2.12. Murine Nasal Colonization and Phage Rescue Experiments

For in vivo phage prevention and rescue experiments, 6–7-week-old (16–22 g) female BALB/c mice were used. Phage suspensions were centrifuged (15,000 × *g*, 30 min) and washed twice in DW. The resulting suspension had a titer of 2.6 × 10^9^ PFU/mL. Logarithmic *K. pneumoniae* 52145 WT cultures were centrifuged, and the pellets were washed twice in phosphate-buffered saline (PBS). OD_600_ was set to 10 to gain a final suspension of 2 × 10^10^ CFU/mL.

In vivo experiments were performed as previously described [30] with some modifications. Animals were cared for in accordance with the guidelines of the European Federation for Laboratory Animal Science Associations (FELASA), and all procedures, care, and handling of the animals were approved by the Animal Welfare Committee of University of Pécs (permit number: BA02/2000-37/2015). Mice were anesthetized with COMBO solution (1:1:0.015 *v*/*v*/*v* physiological saline/calypsol (Richter Gedeon, Budapest, Hungary)/2% Primazin (Alfasan, Woerden, Dutchland)). Seven groups (4 mice per group) were used. All groups were treated with *K. pneumoniae* 52145 WT by injecting 15 µL of OD_600_ = 10 (2.5 × 10^8^ CFU/mice) into the nose using a sterile pipette. Six groups were treated with B1 phage by administering 30 µL of 2.6 × 10^9^ PFU/mL suspension (7.8 × 10^7^ PFU/mice) into the nose at different time points pre/post-infection (−1 h, 1 h, 3 h, 6 h, 12 h, 24 h). One group was not treated with the phage and served as a positive control. General conditions (mass and vitality) and survival of the mice were monitored for 16 days. After 16 days, surviving mice were euthanized by cervical dislocation. Nasal cavity, brain, lungs, spleen, and kidneys were aseptically removed, and tissue homogenates were processed for bacterial and phage presence verification.

## 3. Results

### 3.1. Morphological Features of Phage B1

Four bacteriophages with strong lytic characteristics were purified from wastewater against the WT 52145 *K. pneumoniae* strain. On the basis of their similar restriction patterns with *Eco*RI and *Hind*III, we chose phage B1 for evaluation. Subsequently, during the isolation of a phage against the capsule mutant of the 52145 *K. pneumoniae* strain from a pool of 20 selected and amplified plaques, four phages that were unable to cause lysis on the WT strain were isolated and purified. One (731) was used in our study.

According to TEM analysis, phage B1 had a ≈50–60 nm head and a 150–200 nm long non-contractile, flexible tail, which is a typical feature of the *Siphoviridae* family of the tailed phages (Figure 1a, top).

The phage formed 1.5–2.5 mm diameter individual plaques on its own host bacterial lawn. Around the clear B1 plaques, relatively large (6–10 mm diameter) turbid rings (halo zones) were observed after ON incubation (Figure 1a, bottom). This was a result of differential migration of diffused soluble depolymerases. Halos expanded over time, while the plaques remained the same size.

### 3.2. Growth Characteristics of Phage B1

The one-step growth experiment was performed to determine the latent period and the burst size of the phage on its own host. A triphasic curve was obtained with a latent, a log/rise, and a plateau period (Figure 1b). The B1 phage had a 9 min latency and a relatively steep rise period starting at ≈12 min, and the plateau level was reached at 20 min. Burst size was ≈2200 phage particles per infected bacteria. Adsorption efficiency of phage B1 on its own host was 99.2%.

B1 phage inhibited the growth of *K. pneumoniae* 52145 at different concentrations (Figure 2). The growth of the bacteria was repressed successfully by the phage at MOI between 0.01 and 10,000 in LB broth for 24 h.

### 3.3. Host Spectrum and EOP

Spot tests and PFU determinations were performed with 24 *K. pneumoniae* isolates to measure the host range of phage B1. Results showed that from 19 different capsular serotypes and 3 additional CPS or LPS mutants, the phage was relatively specific for its own serotype. B1 phage was specific to strains with K2 capsule, forming clear plaques and exhibiting high efficiency (EOP = 1) only against its WT host and the identical 52145 from a different source. The effect did not decrease on the LPS mutant strain, and wide halo zones were formed around the clear plaques.

However, when dropped on the CIP 52.217 strain with K13 capsule (a serotype that has a very similar molecular structure to serotype K2), only a turbid spot but no plaque was visible. This indicated capsule depolymerase activity with no lysis. This morphology was described by observing the size growth of the spot over time and spotting the diluted phage suspension. Spots caused by depolymerase activity expanded over time, similar to the halo zone around a clear phage plaque. Upon dilution, spots typically showed reduced transparency and growth.

B1 did not only cause clear lytic plaques on strains with K2 capsule, but also on the double mutant (52145-Δ*wca*_K2_Δ*waaL*), which lacked both capsule and LPS antigen, though the lytic efficiency was lower.

### 3.4. Phage Resistance

Appearance of resistant colonies was observed in the clear plaques of B1 phage. MALDI-TOF MS showed that all the colonies were *K. pneumoniae*. The mutation rate of phage B1 was 1.1 × 10^−3^ per 24 h. However, all 40 selected B1 resistant colonies were sensitive to phage 731, indicating that resistance was acquired through loss of capsule.

### 3.5. Adaptation Limits of Phage B1 to Heat and Detergents

Heat and detergent tolerance of B1 phage was examined. The phage endured incubation at 50 and 60 °C for 1 h without losing its titer (Figure 3a). In the long term, 42 and 37 °C incubations affected the phage with a gradual drop in the titer, completely losing it after 9 months. Although lytic plaques were not detectable by the end of the tested period, the depolymerase activity (translucent spot) was still observable. Phage B1 remained more active at 23 and −20 °C, as titer only slightly dropped to 10^7^ and 10^8^ PFU/mL after 9 months, respectively (Figure 3b). The phage plaques stored at −20 °C in frozen agar did not change their initial titer from 10^7^ PFU/mL after 5 or 18 days, supporting that this storage technique is appropriate for B1 phage.

Concentrated detergents, however, affected the titer drastically. SDS (10%) instantly destroyed phage infectivity and the samples precipitated. The other detergents also reduced phage PFU after 9 months of storage (Figure 3c). Even DW enhanced a major titer drop, suggesting that B1 phage requires a nutrient-rich medium, such as LB, for long-term storage when it is not frozen.

### 3.6. Genomic and Phylogenetic Properties of B1 Phage

Basic genomic properties and sequence identities of the B1 phage were mapped. Basic genome statistics revealed that B1 phage had 30.92 MDa double-stranded, linear DNA with a length of 50,040 base pairs and a G + C content of 50.4% (23.4% adenine, 26.2% cytosine, 24.2% guanine, 26.2% thymine). The genome of B1 was found to contain 77 open reading frames (ORFs), out of which 18 are encoded on the positive and 59 on the negative strand. The average gene length was 588 bp. No remnants of mobile genetic elements and traces of truncated genes were observed.

Databank homologies of *Klebsiella* phage B1 are listed in Appendix A. Besides the listed 14, B1 phage showed high similarity with nearly 50 *Klebsiella* phages, such as GH-K3 (NC_048162.1), 1513 (KP658157.1), KLPN1 (KR262148.1), PKP126 (KR269719.1), and 13 (NC_049844.1). These are all classified as members of the *Webervirus* genus, which was previously enrolled under *Tunavirinae* subfamily, *Siphoviridae* family in the *Caudovirales* order, but according to the recent ICTV taxonomy release (July 2019), the genus is now in the *Drexlerviridae* family, in *Caudovirales* order.

Figure 4 shows the phylogenomic relations of the B1 with other *Klebsiella* phages and reveals the closest relation to *Klebsiella* phage KP36 (JF501022.1).

The comparison of the B1 genome annotations with those of other *Klebsiella* phages was performed (Figure 5), mainly for the identification of a putative depolymerase coding ORF in the B1 genome. These *Klebsiella* phages were previously described as active against the K2 serotype, and their depolymerases were identified. One phage, KP36, was also involved, which is non-K2 specific, but the type species of the *Webervirus* genus and the closest relative of phage B1.

GH-K3 and KP36 belong to the same genus as B1 (Table 2) and have coverage of 93% and 87% with B1, respectively (seq. identity 95.67% and 96.9%, respectively). The comparison reveals that the overall high similarity is impaired only at a definable gene region (Figure 5, striped arrows) between B1 and KP36, and KP36 and GH-K3, but not between B1 and GH-K3. The proteins of the described phages coded by these ORFs have depolymerase activity. Therefore, this 2724 bp long *orf61* of B1 phage was assumed to be a putative depolymerase and was selected to be amplified for further investigations.

Two other *Klebsiella* phages from two different genera and specific for K2 serotype, KpV74 and KpV763, were compared with the previous three phages, but these showed no significant homology with the other three or with each other.

### 3.7. Molecular Properties of the B1 Phage Depolymerase, B1dep

The putative depolymerase coding gene of B1 phage was also characterized. *Orf61* is 2724 bp long and codes for a 907 aa protein, annotated as “phage tail fibers”, which was named “B1dep” after successful expression and purification. It showed strong similarity with depolymerases of *Klebsiella* phage LF20 (99.78%, QQO89608.1), GH-K3 (99.23%, YP_009820105.1), and MMBB (98.79%, QOQ37802.1). Moreover, B1dep had overall low coverage (≈20–30%) with hypothetical and tail fiber proteins of homologous *Klebsiella* phages in the *Webervirus* genus, but similarity was observed in the N-terminal region (first 150–300 residues), which is conserved among these proteins. A conserved phage_tailspike_middle domain was reported by NCBI BLASTp between residues 199–330. A pectin lyase fold was also predicted by InterProScan between amino acids 300–476. This region showed similarity with various lyases and hydrolases according to the HHpred search.

### 3.8. Depolymerase Activity of B1dep

Depolymerase activity of the recombinant protein was detected when spotted on K2 lawn (Figure 6a,b). B1dep formed turbid spots on 52145 WT. This indicates that capsule degradation occurred in the absence of lysis. The spots significantly expanded over time (just like the halo zones around phage plaques), and diluted suspensions (10^−2^) created homogeneous spots with higher opacity, smaller diameter, and slower expansion rate. No spots were observed with control samples: KRX without plasmid or with empty pRSET A.

Activity was observed from the supernatant of the ON-incubated KRX culture, even when the cells were not sonicated before spot testing. Sonicated samples, which did not contain any viable KRX cells, were stored in 500 µL aliquots of sonication buffer at −20 °C and exerted an effect even after years, as seen by the appearance of spots with similar turbidity, expanding over time. However, when the purified B1dep protein samples were stored at −20 °C, they lost their efficiency over time. Boiling at 100 °C for 5 min also eliminated enzymatic activity. In contrast, purified samples stored at 4 °C kept their activity.

Activity of B1dep was also observable instantly when the recombinant protein was spot tested on a 1-day-old 52145 lawn (10^7^ CFU plated). The degradation of the capsule layer was visually detectable after 1–2 min at room temperature (Figure 6a,b). B1 phage (10^7^ PFU/10 µL drop) was also spotted on a grown bacterial lawn and resulted only in a turbid spot even after 1 day incubation on 37 °C.

B1dep had an identical spectrum with B1 phage halos (Table 1), causing a turbid spot without plaques. LPS mutant 52145-Δ*waaL* and strain CIP 52.217 with K13 capsule were also susceptible—this latter case confirmed that the depolymerase of B1 can degrade K13 capsule, but the phage itself is unable to infect and lyse the cells of 52.217 strain, as it was described in Section 3.3. No clearing effect was detected if the depolymerase was applied on the capsule deficient mutant 52145-Δ*wca*_K2_.

### 3.9. Murine Nasal Colonization and Phage Rescue Model

Therapeutic potential of phage B1 was revealed in a murine nasal cavity colonization model. The effect of the phage strongly depended on the time that had passed between bacterial infection and phage administration (Figure 7).

After 24 h of *K. pneumoniae* 52145 infection in mice, the survival rate was 100%. However, the general condition of every group member drastically deteriorated, except for the mice in the group where the B1 phage was administered 1 h prior to the bacterial challenge. The mice in the positive control group (four mice) died within 3 days (LD_50_ = 36 h), and the survival rate drastically dropped in the other where the phage suspension was injected post-infection. All these mice died within 96 h. Survival rate of the pre-treated group (Figure 7, B1 phage −1 h) was 100% (four mice) after 9 days and dropped to 50% by day 13. Sixteen days after bacterial challenge, the two surviving mice (m3 and m4) were sacrificed. CFU determination showed that the nasal cavity, lungs, spleen, kidneys, and the brain contained bacteria in relatively high numbers (3 × 10^3–4^ CFU/organ). The nasal cavity was colonized by the 52145 strain, which caused a systemic infection, and the bacteria were still present in the distant organs of the surviving mice.

No phage activity was observed in any of the processed organs.

## 4. Discussion

To date, only a few phages have been described as specifically acting against *Klebsiella pneumoniae* strains with the virulent K2 serotype. Due to its resistance and intracellular killing, this serotype, along with K1, is the most frequently isolated serotype collected from patients worldwide [5,10,12,13,14,15,16,17], with a particularly high prevalence in liver abscesses and endophthalmitis [18]. The spread of emerging antibiotic resistance among the isolates requires novel methods to be seriously considered as potential therapeutic or preventive agents.

The target microorganism in our study was the hypervirulent *Klebsiella pneumoniae* strain, 52145 [51], a reference strain of serotype O1:K2 that has been thoroughly analyzed since its first isolation [2,56,57]. This strain belonged to the ST66-K2 lineage [17], which was recently reported to cause community-acquired infections [58].

The low rate of resistance against the lytic activity of B1 was a favorable feature, with minimal resistant colonies occurring at MOI = 0.01. Complete efficacy (100%) of phage 731 on every tested phage-resistant colony (40/40) and their MALDI-TOF-based species confirmation clearly indicated that all phage-resistant colonies lost their capsules. The therapeutic importance of this observation is that capsule-deficient mutants typically lose their virulence and become sensitive to phagocytosis [59], a typical feature that characterizes the K2 serotype of *K. pneumoniae* [18]. It was revealed in our previous pilot experiment that the capsule-less 52145 mutant did not cause death in a murine lung infection model (Appendix A).

Lack of capsule-free variants of 52145 in the organ homogenates of the two sacrificed mice on the 16th day post-challenge supported the observations that capsule is indispensable for in vivo survival of *K. pneumoniae* [60,61]. The lack of bacteriophages in the tissue homogenate suggested that phage B1 either (i) was not translocated into the circulatory system, which is contraindicated by the extended lifespan of the mice, or (ii) did enter the circulation, but its level dropped under the detection limit in the two-week time interval. Although passive phage translocation throughout the body can occur [62,63], we hypothesize that B1 phage virions were actively translocated through the mucous epithelial layer through the infection of 52145, and by this, they were able to prolong the life of some mice. We have previously reported the biodistribution of a K1 *E. coli* phage after its use as a therapeutic intervention [64], but we could not show this effect in case of B1.

Our results demonstrated the protective effect of B1 if it is administered 1 h prior to bacterial challenge (B1 phage −1 h). This is in accordance with recent findings that mucus-adhered bacteriophages provide non-host-derived immunity [65]. The limited therapeutic potential of phage B1 could be because 1 h was sufficient for 52145 to gain marked advantage in this rapid infection model and to establish a fatal systemic infection.

Narrow specificity of the newly isolated *Klebsiella* phage B1 to the K2 serotype was not unexpected. This is a typical feature of *Klebsiella pneumoniae* bacteriophages and mostly determined by the depolymerase that they carry. The polyvalence of particular phages is due to the fact that they are equipped with more depolymerases, and accordingly, their broad lytic spectrum matches the specificity range of their individual depolymerases [66,67,68]. Recently, a phage was described with two depolymerases, both acting against K47, but to different subsets of strains. This was due to the subtle differences in the sugar chain composition of the K47 capsules, showing that depolymerase specificity can be even more delicate than the capsule serotype level [69].

The elucidation of a gene region (*orf61*) in B1 with suggested depolymerase activity was first performed by predictions. Identification of a tail fiber with putative depolymerase motif was a strong hint for the localization of the capsule depolymerase, as depolymerases are typically associated with tail fibers [70,71]. Limited gene bank data about capsule depolymerase proteins were available, and only a couple of studies have dealt with cloning and characterization of such enzymes specific for KN2, K1, or K63 [72,73,74]. Although the list of published *Klebsiella* phage depolymerases has grown [36], to the best of our knowledge, few phages and/or depolymerases have been described as specifically acting against *Klebsiella pneumoniae* strains with the virulent K2 serotype (Table 2). Although expression of one depolymerase (Dep_kpv74) from a K2-specific phage (KpV74) was recently reported, no homology between *kpV74_56* (Dep_kpv74) and *orf61* (B1dep) of B1 could be revealed (Figure 5). The reason for the lack of homology between them could be that while B1 is one member of the *Siphoviridae* family, while KpV74 is one representative of the *Podoviridae* family [37]. Homology of KpV74 to KpV763 (another K2-specific *Podoviridae* member), reported previously [37], was also low. This is a strong hint for their convergent evolutionary process, which demonstrated that capsule specificity of the depolymerase is not necessarily in coherence with phage relatedness. Phage GH-K3 is a siphophage and the only member of the same genus as B1 with K2 depolymerase activity [75]. The sequence coded by gene *gp32* is also highly similar with B1dep (99.23%). However, this protein was not yet expressed and characterized functionally, only predicted as a depolymerase. Two more depolymerases of phages LF20 (MW417503.1) and MMBB (MT894005.1) showed this level of similarity, also from *Webervirus* genus, although no literature is available regarding these.

The characteristic clearing zone [32,76] on the lawn of 52145 caused by depolymerase activity after dropping of the purified B1dep enzyme clearly indicated that *orf61* was the functional depolymerase gene, making this the first reported expressed depolymerase encoded by a K2 serotype specific bacteriophage, belonging to the *Siphoviridae* family.

Durability of phage B1 under different physical conditions including temperature showed similarity with other published members of the *Webervirus* genus [24,27,77]. However, B1 had slightly better durability at 37, 40, and 42 °C, also when compared to a *Klebsiella* podophage [28]. A visible decrease in the B1 phage titer after 60 min only appeared at 70 °C. Long-term exposure of phages to concentrated detergents was not examined previously, although it could be a relevant aspect in bacteriophage-based prevention or disinfection. Although B1 phage lost its activity after 9 months in all the detergents, gradual loss of infectivity suggests that the phage is not instantly disabled in these conditions. It is active after a few days of mixing, and thus it may maintain prolonged activity in diluted detergents. These traits could be beneficial for phage-based therapy.

## 5. Conclusions

Our results suggest that the B1 phage and its capsule depolymerase, B1dep, can be promising candidates for further investigation in phage-therapy research. As this phage targeted a hypervirulent serotype, and all its examined properties were suitable, our results may aid the development of bacteriophage-based therapeutic strategies for *Klebsiella pneumoniae* infections, specifically targeting hypervirulent strains.

## Figures and Tables

**Figure 1 microorganisms-09-00650-f001:**
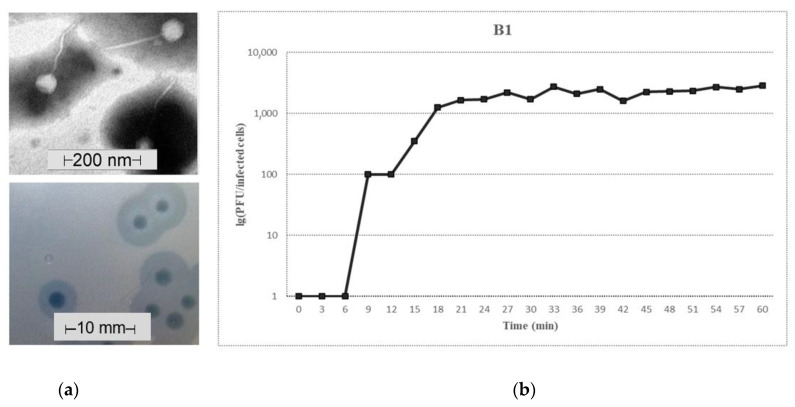
Characteristics of the B1 phage. (**a**) Electron micrograph of B1 phage stained with 1.5 *w*/*v* phospho-tungstic acid (**top**) and plaques on 52145 WT lawn (**bottom**). Scale bars represent 200 nm and 10 mm, respectively. Individual plaques were acquired by agar overlay method. (**b**) One-step growth curve of phage B1. The plaque-forming unit (PFU) per infected cell at different times are shown. Data are the mean of three independent experiments.

**Figure 2 microorganisms-09-00650-f002:**
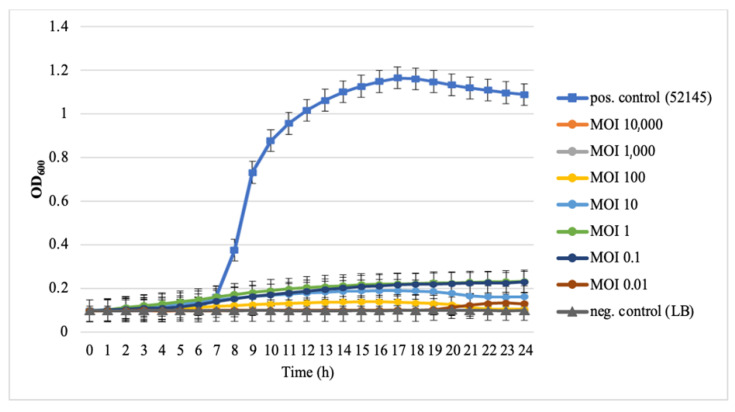
Time-kill assay of phage B1 against *Klebsiella pneumoniae* 52145. Overnight bacterial culture was diluted (10^3^ CFU/mL) and infected with phage B1 at different multiplications of infection (MOIs). Non-phage-treated bacterial cultures acted as a positive control (■), lysogeny broth (LB) without bacteria acted as a negative control (▲). Each data point is the mean from three experiments. Standard deviations (±SD) are shown as vertical lines.

**Figure 3 microorganisms-09-00650-f003:**
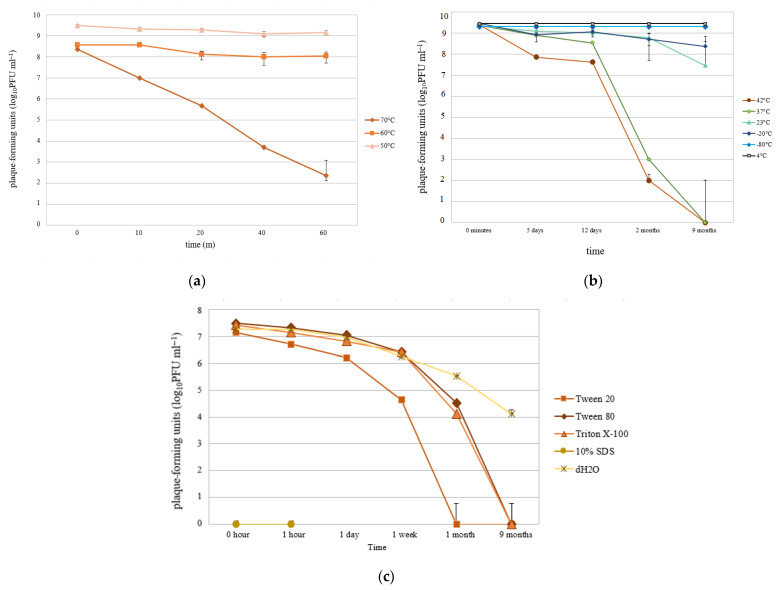
Adaptation limits of phage B1. (**a**,**b**) Heat tolerance. (**c**) Detergent tolerance. Data are presented in PFU by time. Each data point is the mean from three experiments. Standard deviations (±SD) are shown as vertical lines.

**Figure 4 microorganisms-09-00650-f004:**
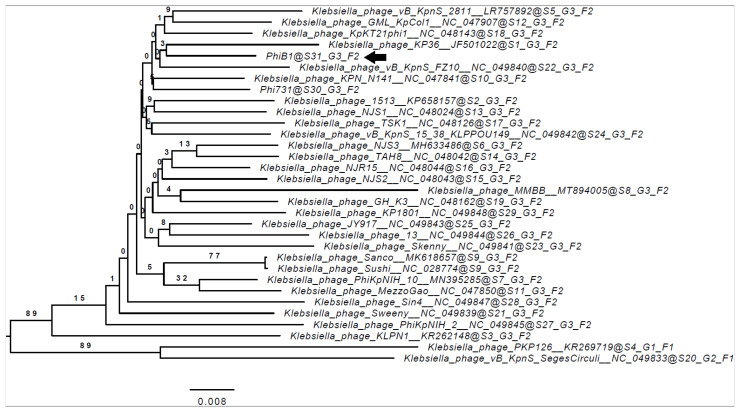
Whole genome-based phylogenetic relations of phage B1. Genome–BLAST distance phylogeny (GBDP) trees were inferred using the formula D4 and yielded average support of 9%. The numbers above branches are GBDP pseudo-bootstrap support values from 100 replications. The branch lengths of the resulting VICTOR trees are scaled in terms of the respective distance formula used. The OPTSIL clustering (results indicated after the @ mark) yielded 31 species clusters (S), 3 clusters at the genus level (G), and 2 at the family level (F). Accession numbers are also indicated next to the phage names. *Klebsiella* phage B1 is marked with a black arrow (
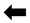
).

**Figure 5 microorganisms-09-00650-f005:**
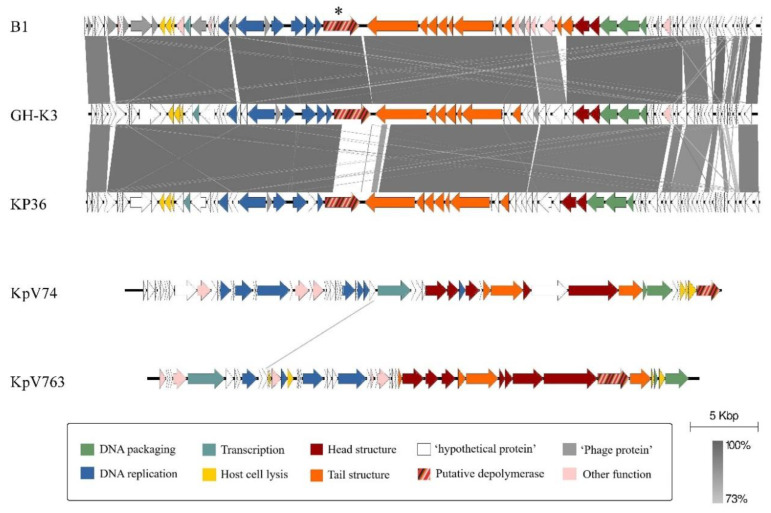
Representation of the homology similarities and differences of *K. pneumoniae* K2 serotype-specific phages and one reference phage of the *Webervirus* genus. Phage B1 (K2 specific, *Siphoviridae*), GH-K3 (K2 specific, *Siphoviridae*), KP36 (K63 specific, *Siphoviridae*, reference phage of the *Webervirus* genus), KpV74 (K2 specific, *Podoviridae*), and KpV763 (K2 specific, *Podoviridae*). *Orf61* of phage B1 is marked with an asterisk (*****). Homology was only restricted to the N-terminal between the depolymerase genes of B1 and KP36, as well as between GH-K3 and KP36, while no homology could be revealed to the two K2 specific podophage sequences. Annotations of the genomes were performed by the RAST server and were visualized by Easyfig.

**Figure 6 microorganisms-09-00650-f006:**
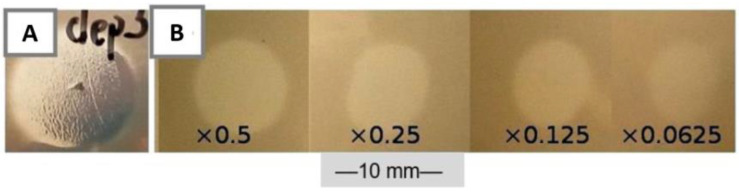
Capsule-degrading activity of B1dep. Spot tests of the purified recombinant B1dep protein on 52145 WT lawn. (**A**) Turbid spot of concentrated crude KRX lysate, resembling the plaque halo. (**B**) Spots of dilution series of purified B1dep. A total of 10 µL was spotted from every dilution. Scale bar represents 10 mm.

**Figure 7 microorganisms-09-00650-f007:**
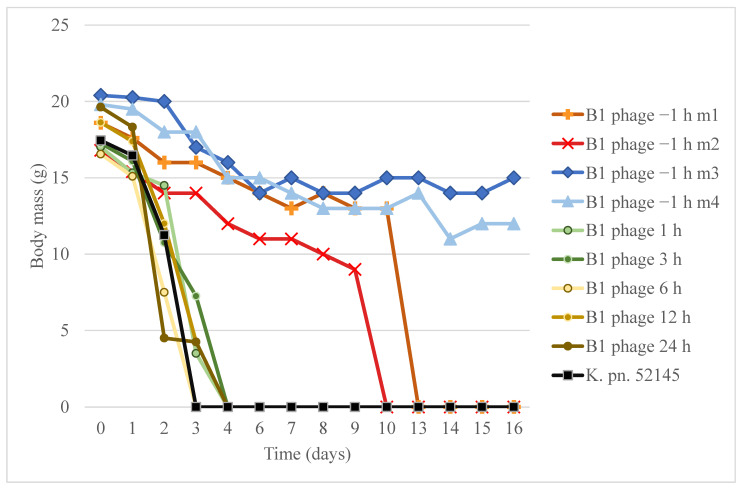
Comparison the therapeutic and preventive effect of B1 phage in frame of survival and bodyweight in a murine nasal infection model. The graph shows body mass of mice from the start of nasal cavity infection over 16 days. Body mass in groups of therapeutic phage administrations (B1 phage 1 h, 3 h, 6 h, 12 h, 24 h) and in the *K. pneumoniae* 52145 infection positive control (K. pn. 52145) are shown as the mean of four mice from the same group. Body mass graphs of mice in the preventive phage administered group (−1 h) are indicated individually (B1phage −1 m1, m2, m3, m4).

**Table 2 microorganisms-09-00650-t002:** Comparison of *Klebsiella* phages with K2 depolymerase activity. The phage genome similarities are presented in Figure 5. The old and recent classification of the phages are indicated, according to the International Committee on Taxonomy of Viruses (ICTV) website (https://talk.ictvonline.org/taxonomy/, access date: 19 March 2021), along with capsule specificity and recent research status of each depolymerase.

*Klebsiella* Phage	Old Classification (ICTV, 2015–2016)	Recent Classification (ICTV, July 2019)	Capsule Specificity	Depolymerase Gene and Recomb. Protein Name	Depolymerase Status
B1	*Siphoviridae, Tunavirinae*, *Kp36virus*	*Drexlerviridae, Tunavirinae*, *Webervirus*	K2	*orf61*, B1dep	expressed
GH-K3			K2	*gp32*	predicted
KP36			K63	*gp50*, DepoKP36	expressed
KpV74	*Podoviridae, Autographivirinae*, *Kp34virus*	*Autographiviridae, Slopekvirinae*, *Drulisvirus*	K2	*kpv_77_56*, Dep_kpv74	expressed
KpV763	*Podoviridae, Autographivirinae*, *Kp32virus*	*Autographiviridae, Studiervirinae,* *Przondovirus*	K2	*kpv_763_43*	predicted

## Data Availability

All data are presented in this manuscript in the main text and in the Appendix A.

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
