# Peer review of "Isolation and Characterization of a Novel Lytic Bacteriophage against the K2 Capsule-Expressing Hypervirulent *Klebsiella pneumoniae* Strain 52145, and Identification of Its Functional Depolymerase"

_microorganisms, 2021, doi:10.3390/microorganisms9030650_

Round 1

Reviewer 1 Report

In the manuscript “Isolation and characterization of a novel lytic bacteriophage against the K2 capsule-expressing 3hypervirulent Klebsiella pneumoniae strain 52145, and identification of its functional depolymerase” Botond Zsombor Pertics and colleagues reported on a bacteriophage against the nosocomial K. pneumoniae K2 capsular serotype. For the phage a narrow host range and a stable lytic activity was described. The Klebsiella phage B1 belongs to the Webervirus genus in Drexlerviridae family. Further phenotypic and genotypic properties were described.

It is a very nicely paper on Klebsiella phages, which is well written and carefully conducted. Some of the below requested information need to be added. If experiments show in figure 2, 3 and 7 were not conducted in triplicate, the strongness of the suggestions given in the text need to be alleviated. Please check carefully…

Nice Paper!

LB is not Luria Bertani medium – it is lysogeny broth – if you can not believe please read the review of Bertani itself! Please change

A meaningful reference for MALDI investigation needs to be given… especially for the method used!

Please provide detailed information on “mid-log suspension of isolate 5214“” what does this mean – optical density values are more reliable

“phage B1 has a ~50-60 nm head and a 150-200 nm” how many phage particles were measured …

Can you also provide the number of measured phages ten ore more are appropriate…

Figure 2, 3 and 7: was the experiment conducted in triplicate… please provide standard deviation, if possible…

Table 1: Please specify EOP terms ineffective, high… for better assessment

Author Response

Detailed Responses to Editor and Reviewers

Manuscript Number: microorganisms-1146689

Response to the Reviewers’ comments

We are very pleased to resubmit for publication the revised version of Isolation and characterization of a novel lytic bacteriophage against the K2 capsule-expressing hypervirulent Klebsiella pneumoniae strain 52145, and identification of its functional depolymerase” (microorganisms-1146689) by Botond Zsombor Pertics, Alysia Cox, Adrienn Nyúl, Nóra Szamek, Tamás Kovács and György Schneider to be considered for publication as an original article in Microorganisms. We carefully considered your comments, hoping our revision has improved the paper to a level of your satisfaction.

Again, we appreciate the opportunity to revise our work for consideration for publication in Microorganisms.

The authors thank the reviewers for their rapid and constructive reviews of the manuscript. Here are our detailed responses to the reviewers' issues. Reviewers' comments are reported in red.

Reviewer 1

In the manuscript “Isolation and characterization of a novel lytic bacteriophage against the K2 capsule-expressing 3hypervirulent Klebsiella pneumoniae strain 52145, and identification of its functional depolymerase” Botond Zsombor Pertics and colleagues reported on a bacteriophage against the nosocomial K. pneumoniae K2 capsular serotype. For the phage a narrow host range and a stable lytic activity was described. The Klebsiella phage B1 belongs to the Webervirus genus in Drexlerviridae family. Further phenotypic and genotypic properties were described.

It is a very nicely paper on Klebsiella phages, which is well written and carefully conducted. Some of the below requested information need to be added. If experiments show in figure 2, 3 and 7 were not conducted in triplicate, the strongness of the suggestions given in the text need to be alleviated. Please check carefully…

Nice Paper!

LB is not Luria Bertani medium – it is lysogeny broth – if you can not believe please read the review of Bertani itself! Please change

 Response: Thank you for your comment. The abbreviation was modified accordingly.

A meaningful reference for MALDI investigation needs to be given… especially for the method used!

  Response: Thank you for your remark. A reference is now added to the paragraph, in which the MALDI investigation method is described in details (section 2.4, line 151-155).

Please provide detailed information on “mid-log suspension of isolate 5214“” what does this mean – optical density values are more reliable

  Response: Thank you for your remark. The mid-log phase was determined as OD600=0.5-0.6, but it was indeed not consistent throughout the text. It is now modified, with the explanation of the abbreviation OD (optical density) (lines 105-106).

“phage B1 has a ~50-60 nm head and a 150-200 nm” how many phage particles were measured …

Can you also provide the number of measured phages ten ore more are appropriate…  

 Response: At least 20 clear and compete phage particles were captured by electron microscopy. These were compared to each other and showed similar morphology, in terms of size and shape. The size of the capsid head and tail was determined with the scale bar (200 nm) indicated on Fig.1a, which was applied on the picture by the software of the TEM and was represented on the figure without modifications.

The size of the phage plaques and halos were determined with a ruler. Many plaques (n>30) and their halos were measured on the agar after ON incubation (Fig. 1b). The scale bar was put on the figure manually (based on the style of the one above), but it is proportional to the size of the petri dish/plaques/halos and represents valid linear units.

Figure 2, 3 and 7: was the experiment conducted in triplicate… please provide standard deviation, if possible…

 Response: Thank you for your observation. On Figure 2, error bars are now present, and the figure description has been expanded with the following sentence: “Each data point is the mean from 3 experiments. Standard deviations (±SD) are shown as vertical lines” (line 414-415). Section 2.5 was also complemented: “Experiments were performed in three replicates and reproduced in three independent trials” (lines 200-201), as this statement was not written previously.

Error bars are added to Figure 3 as well, and the following sentences were written to the description: “Data are presented in PFU by time. Each data point is the mean from 3 experiments. Standard deviations (±SD) are shown as vertical lines” (line 480-481)

We did not put error bars on Figure 7, because the experiment was not performed in triplicates. Although there were 4 mice in every group, and their mean body mass values were indicated on the graph (except for the preventive phage treated group where the 4 mice were plotted individually), this experiment was not replicated at a different time. We wanted to calibrate the preventive and/or therapeutic efficacy of the phage with this experiment. The results clearly showed that after 4 days, all mice died unanimously that were treated with the phage post-infection. Following the 3R strategy, reduction of used animals was kept in mind, and the results clearly indicated that there is no need to repeat the experiment with the same conditions. Of course, further studies are under planning, but with the exclusion of groups 3 h, 6 h, 12 h, 24 h.

Table 1: Please specify EOP terms ineffective, high… for better assessment

 Response: Thank you for your remark. The specifications of possible EOP values are now written in the description of table 1 (lines 448-450), copied from the corresponding Materials and Methods section (2.3).

Reviewer 2 Report

The authors present an interesting paper describing the characterisation of a novel phage specific to K. pneumoniae of the K2 serotype and the identification of its functional polymerase. I don't have any major concerns with the science or the interpretation of results. The majority of the paper is well written however, I did find the results section confusing and a little difficult to follow at times. Please see below for specific comments/suggestions:

  • Line 62-63 - combine this with the paragraph beneath it.
  • Line 66 - change "bacterium" to "bacterial".
  • Line 82 - change "therapeutical" to "therapeutic".
  • Section 2.1 - a little more information regarding the origins of your novel strains is required. Were these newly isolated clinical isolates? Is this why the MALDI-Tof was necessary?
  • Line 117 - should "10ml" read "10μl"?
  • Line 120-122 - I would consider rephrasing this sentence. 
  • Section 2.3 - you state three descriptions of possible outcomes however, your table 1 indicates 4 possible outcomes. Halos can be present both with or without lysis.
  • Line 151 - remove the word "respectively".
  • Line 360 - change "survived" to "surviving".
  • Line 367-368 - I don't understand the relevance of EcoRI and HindIII restriction patterns and the selection of phage B1. 
  • Line 377 - should "6-10nm" read "6-10mm"?
  • Lines 420-424 - perhaps mention that this was only observed in one Kp isolate.
  • Lines 425-432 - I would remove this from here as this result is covered later in section 3.8. 
  • Figure 3c - change the line colours. Different shades of yellow are not easily discernible and 10% SDS and dH2O can barely be seen.
  • Line 475 - change "Table II" to "Table 2".
  • Line 490 - I don't understand the reference to Table 1 here. Should this be Table 3? The sentence is also a little confusing as it is. Consider changing to "......belong to the same genus as B1 and have coverage of ........"
  • Lines 498-501 - it is not clear what the purpose of this comparison is. A sentence beforehand stating the reasoning behind this would be helpful to the reader.
  • Table 2 - move to the supplementary section.
  • Figure 4 - I can't see the black arrow indicating the location of B1. If it is present, it needs to be made more obvious.
  • Table 3 - there is no reference to it in the text currently, though I assume it has mistakenly been referred to as Table 1 on line 490. Consider moving to supplementary section.
  • Lines 552-558 - this paragraph is confusing and I don't understand its purpose. I don't feel it add much to the paper and could be omitted. 

General comments:

  • the results section could benefit from being presented in a different order. Also, a sentence clarifying the purpose of the experiment at the beginning could make for easier reading in some sections. 
  • Lines 629-623 - have the authors considered whether systemic delivery of phage (either via IP or IV) could lead to better dissemination of phage and a better survival outcome?
  • Do the authors know if loss of capsule leads to sensitisation to antibiotics? If yes, it might be worth mentioning briefly in the discussion. Alternatively, it may be worth investigating in future studies.

Author Response

Detailed Responses to Editor and Reviewers

Manuscript Number: microorganisms-1146689

Response to the Reviewers’ comments

We are very pleased to resubmit for publication the revised version of Isolation and characterization of a novel lytic bacteriophage against the K2 capsule-expressing hypervirulent Klebsiella pneumoniae strain 52145, and identification of its functional depolymerase” (microorganisms-1146689) by Botond Zsombor Pertics, Alysia Cox, Adrienn Nyúl, Nóra Szamek, Tamás Kovács and György Schneider to be considered for publication as an original article in Microorganisms. We carefully considered your comments, hoping our revision has improved the paper to a level of your satisfaction.

Again, we appreciate the opportunity to revise our work for consideration for publication in Microorganisms.

The authors thank the reviewers for their rapid and constructive reviews of the manuscript. Here are our detailed responses to the reviewers' issues. Reviewers' comments are reported in red.

Reviewer 2.

The authors present an interesting paper describing the characterisation of a novel phage specific to K. pneumoniae of the K2 serotype and the identification of its functional polymerase. I don't have any major concerns with the science or the interpretation of results. The majority of the paper is well written however, I did find the results section confusing and a little difficult to follow at times. Please see below for specific comments/suggestions:

Line 62-63 - combine this with the paragraph beneath it.

Response: Thank you for your comment. The two paragraphs are now combined.

Line 66 - change "bacterium" to "bacterial".

Response: Thank you, it is changed to “bacterial” cells.

Line 82 - change "therapeutical" to "therapeutic".

Response: Thank you, “therapeutical” is now changed to “therapeutic”.

Section 2.1 - a little more information regarding the origins of your novel strains is required. Were these newly isolated clinical isolates? Is this why the MALDI-Tof was necessary?

Response: Thank you for your remark. The strains were not novel, the reference for each used strain is indicated in Table 1. CIP strains were purchased from the Pasteur Institute, France, the other mentioned ATCC, NTUH and MGH strains were available in our institute. The 52145 strain and its derivatives were provided by Professor Juan Thomas from the University of Barcelona involved in Acknowledgements. MALDI-TOF was necessary to confirm the species of the resistant colonies growing in B1 clearing zones, to make sure that those are Klebsiella pneumoniae colonies, which became resistant and not just contamination. So the whole MALDI-TOF paragraph was transferred from section 2.1 to 2.4. “Phage resistance detection”, where it is more relevant (line 148-152). Results section 3.4. was expanded with the statement: “MALDI-TOF MS showed that all the colonies were K. pneumoniae” (line 440).

Line 117 - should "10ml" read "10μl"?

Response: Thank you, the correct unit is “10 µl” indeed. It is corrected.

Line 120-122 - I would consider rephrasing this sentence. 

Response: Thank you for the remark. The sentence was rephrased: “‘Klebsiella phage 731’ was isolated the same way on the 52145 capsule mutant (ΔK), furthermore it was selected to be ineffective on the wild type (WT)” (line 119-121).

Section 2.3 - you state three descriptions of possible outcomes however, your table 1 indicates 4 possible outcomes. Halos can be present both with or without lysis.

Response: Thank you for your comment. Section 2.3 was supplemented with the following sentence: “Clear spots were observed either sole or surrounded by a turbid ring (halo), this latter indicating lysis and depolymerase activity as well” (line 127-129).

Line 151 - remove the word "respectively".

Response: Thank you. The unnecessary word was deleted.

Line 360 - change "survived" to "surviving".

Response: The word was changed, according to your observation.

Line 367-368 - I don't understand the relevance of EcoRI and HindIII restriction patterns and the selection of phage B1. 

Response: Thank you for your remark. The isolation procedure (3 rounds) resulted 4 phage suspensions (named B1, B2, B3, B4). These were similar according to their morphology by TEM and plaque formation. We wanted to confirm the similarity with a quick method, and to make sure that these four phage suspensions are containing the same phage. For this purpose, we conducted a restriction experiment of the four phage DNA. Restriction patterns were the same for the four phages, so we assumed that they are similar and we selected the first suspension (B1) for further studies, knowing that B2, B3 and B4 are identical and could serve as a reserve. The case was the same with the 731 phage: 731, 732, 733 and 734 were identical, and 731 was used for the experiments.

Line 377 - should "6-10nm" read "6-10mm"?

Response: Thank you. It was only a typo and was corrected.

Lines 420-424 - perhaps mention that this was only observed in one Kp isolate.

AND

Lines 425-432 - I would remove this from here as this result is covered later in section 3.8. 

Response: Thank you for your comments. The 2 paragraphs (3.3 and 3.8) were reorganized and cleared of redundancy. 3.3. now mentions the one strain, 52.217, with K13 capsule, which is not lysed by the B1 phage, but a turbid spot is observed upon spotting of the phage. In section 3.8, the fact, that the depolymerase of B1 can degrade K13 capsule, but the phage itself does not cause lysis, is confirmed and mentioned.

Figure 3c - change the line colours. Different shades of yellow are not easily discernible and 10% SDS and dH2O can barely be seen.

Response: Thank you for your observation. The colors are now changed to brown/orange and dark yellow, this way it is more visible.

Line 475 - change "Table II" to "Table 2".

Response: Thank you. The misspelling was corrected.

Line 490 - I don't understand the reference to Table 1 here. Should this be Table 3? The sentence is also a little confusing as it is. Consider changing to "......belong to the same genus as B1 and have coverage of ........"

Response: Thank you for your remark. The reference was wrong and should be “Table 3”. It is corrected. The sentence was changed as well, according to your suggestion (line 509).

Lines 498-501 - it is not clear what the purpose of this comparison is. A sentence beforehand stating the reasoning behind this would be helpful to the reader.

Response: Thank you for your observation. The following sentence was added before, explaining that the purpose of this comparison was the identification of a depolymerase in B1. „The comparison of the B1 genome annotations with those of other Klebsiella phages was performed (Fig. 5), mainly for the identification of a putative depolymerase coding ORF in the B1 genome. These Klebsiella phages were previously described as active against the K2 serotype, and their depolymerases were identified” (line 502-506).

Table 2 - move to the supplementary section.

Response: Thank you for your remark. The table was moved to the supplementary data section as Table S1, and the former “Table 3” is now changed to “Table 2” in the description and all references in the main text.

Figure 4 - I can't see the black arrow indicating the location of B1. If it is present, it needs to be made more obvious.

Response: Thank you. The missing of the arrow was only a formatting error, as the arrow was drawn with Word individually on the picture and got lost during reformatting the manuscript. The arrow is now added permanently to the figure.

Table 3 - there is no reference to it in the text currently, though I assume it has mistakenly been referred to as Table 1 on line 490. Consider moving to supplementary section.

Response: Thank you for the correction. Indeed, it was a typo, the text should have been referring to Table 3. It is corrected. We decided not to move this table to the supplementary section, as it is an organic explanation/complementation of Figure 5, and the concept is more transparent if the two objects (Fig.5 and Table 3) are adjacent in the text.

Update: The table referred here and in previous responses as “Table 3” is now “Table 2”, as the previous “Table 2” was moved to the supplementary section and became “Table S1”.

Lines 552-558 - this paragraph is confusing and I don't understand its purpose. I don't feel it add much to the paper and could be omitted. 

Response: Thank you for your comment. The statement: “No spots were observed with control samples KRX without plasmid or with empty pRSET A” is important to show that turbid spots on the lawn are not caused by the effect of KRX cell lysate or by some other artifact, but by the recombinant depolymerase protein, coded by the orf61 insert in pRSET A plasmid. Since the depolymerase activity was not demonstrated with other methods yet, it was important to emphasize that this turbid clearance is doubtlessly caused by the B1dep protein.

General comments:

the results section could benefit from being presented in a different order. Also, a sentence clarifying the purpose of the experiment at the beginning could make for easier reading in some sections. 

Response: Thank you for your comment. The following Results sections were expanded with 1-2 sentences on their beginning: 3.3, 3.4, 3.5, 3.6, 3.7. This will probably clarify the purpose of the experiments and will provide a better understanding of the whole concept of this study.

Lines 629-623 - have the authors considered whether systemic delivery of phage (either via IP or IV) could lead to better dissemination of phage and a better survival outcome?

Response: Thank you for the remark. Both IP and IV administration were considered to be probably effective as preventive or therapeutic method of phage administration. First, we aimed to confirm or sort out the effectiveness of nasal delivery route (same location as the bacterial colonization). However, in the future we are planning to administer the phage by the above-mentioned delivery routes as well.  

Do the authors know if loss of capsule leads to sensitisation to antibiotics? If yes, it might be worth mentioning briefly in the discussion. Alternatively, it may be worth investigating in future studies.

Response: Thank you for your observation. It was mentioned in the discussion that capsule loss leads to sensitization to phagocytosis. It is also revealed in this study that capsule deprivation sensitizes the bacteria to a different phage (731), originally not effective against the strain. Sensitization to antibiotics was not yet examined in our laboratory, but it will be tested in the near future. Furthermore, regarding virulence, in one of our previous pilot experiments it was shown that the capsule-less mutant did not cause death in a murine lung infection model. This statement was added to the ominous paragraph (line 639-641), along with a supplementary figure (Figure S1) to the Supplementary material which presents the results of this pilot study.

Round 2

Reviewer 1 Report

The requested revisions were carefully conducted.